# Genome Characterization of Bird-Related Rhabdoviruses Circulating in Africa

**DOI:** 10.3390/v13112168

**Published:** 2021-10-27

**Authors:** Dong-Sheng Luo, Zhi-Jian Zhou, Xing-Yi Ge, Hervé Bourhy, Zheng-Li Shi, Marc Grandadam, Laurent Dacheux

**Affiliations:** 1CAS Key Laboratory of Special Pathogens and Biosafety, Wuhan Institute of Virology, Chinese Academy of Sciences, Wuhan 430071, China; dongshengluo@outlook.com (D.-S.L.); zlshi@wh.iov.cn (Z.-L.S.); 2University of Chinese Academy of Sciences, Beijing 100049, China; 3Lyssavirus Epidemiology and Neuropathology Unit, Institut Pasteur, Université de Paris, 75015 Paris, France; herve.bourhy@pasteur.fr; 4Hunan Provincial Key Laboratory of Medical Virology, College of Biology, Hunan University, Changsha 410082, China; zjzhou@hnu.edu.cn (Z.-J.Z.); xyge@hnu.edu.cn (X.-Y.G.); 5Institut de Recherche Biomédicale des Armées, 91220 Bretigny-sur-Orge, France; marc.grandadam@orange.fr; 6National Reference Center for Arboviruses, Institut Pasteur, Université de Paris, 75015 Paris, France

**Keywords:** rhabdovirus, *Sunrhavirus*, bird, genome, Africa, NGS, genetic diversity, arbovirus

## Abstract

*Rhabdoviridae* is the most diverse family of the negative, single-stranded RNA viruses, which includes 40 ecologically different genera that infect plants, insects, reptiles, fishes, and mammals, including humans, and birds. To date, only a few bird-related rhabdoviruses among the genera *Sunrhavirus*, *Hapavirus*, and *Tupavirus* have been described and analyzed at the molecular level. In this study, we characterized seven additional and previously unclassified rhabdoviruses, which were isolated from various bird species collected in Africa during the 1960s and 1970s. Based on the analysis of their genome sequences obtained by next generation sequencing, we observed a classical genomic structure, with the presence of the five canonical rhabdovirus genes, i.e., nucleoprotein (N), phosphoprotein (P), matrix protein (M), glycoprotein (G), and polymerase (L). In addition, different additional open reading frames which code putative proteins of unknown function were identified, with the common presence of the C and the SH proteins, within the P gene and between the M and G genes, respectively. Genetic comparisons and phylogenetic analysis demonstrated that these seven bird-related rhabdoviruses could be considered as putative new species within the genus *Sunrhavirus*, where they clustered into a single group (named Clade III), a companion to two other groups that encompass mainly insect-related viruses. The results of this study shed light on the high diversity of the rhabdoviruses circulating in birds, mainly in Africa. Their close relationship with other insect-related sunrhaviruses raise questions about their potential role and impact as arboviruses that affect bird communities.

## 1. Introduction

Rhabdoviruses are enveloped RNA viruses belonging to the order *Mononegavirales*. They are characterized by a bullet or rod shape and contain a single or segmented molecule of linear negative-strand RNA of a size approximately 10−16 kb, which contains the five canonical genes encoding the nucleoprotein (N), the phosphoprotein (P), the matrix protein (M), the glycoprotein (G), and the RNA-dependent RNA polymerase (L) [1,2]. Moreover, various novel and diverse accessory genes or putative open reading frames (ORFs) overlap these genes or are interspersed between them [3]. Each gene and some of the accessory ORFs are flanked by relatively well-conserved transcription initiation (TI) and transcription termination polyadenylation (TTP) sequences [1,3].

The family *Rhabdoviridae* is the most diverse within the *Mononegavirales*, with 40 different genera and 246 species according to the latest update by the International Committee on Taxonomy of Viruses (ICTV: https://talk.ictvonline.org/, accessed on 25 September 2021) [2]. The members of this family exhibit a large ecological diversity, with pathogens infecting various plants or animals, including mammals, such as livestock and humans, insects, fishes, reptiles, and birds. Some of them have significant public health, livestock, aquaculture, and agricultural impacts [4]. However, dozens of putative or unclassified new species are waiting to be assigned to potential new genera in the near future. Early taxonomy of these viruses was based on virion morphology and serological cross-reactivity. Thus, many unclassified rhabdoviruses had been assigned to certain taxa, according to serological cross-reactivity with some typical members of the same rhabdovirus genus or group. For instance, Tupaia virus (TUPV) and Klamath virus (KLAV) had been related to *Vesiculovirus* genus [5,6,7], but subsequent molecular analysis definitively classified them into the new and distinct *Tupavirus* genus [1]. Similarly, the previously uncharacterized Duvenhage lyssavirus (DUVV), Lagos bat lyssavirus (LBV), Mokola lyssavirus (MOKV), European bat lyssavirus 1 (EBLV-1), and European bat lyssavirus 2 (EBLV-2) were initially classified into rabies-related viruses by serological test before being considered as individual species among the genus *Lyssavirus* [8]. As the development of molecular and sequencing techniques have become more and more efficient and available, genotyping is now considered to be a key element in viral taxonomy [9,10].

Until now, 12 bird-related viruses were encompassed within the family *Rhabdoviridae*, in three different genera, namely *Tupavirus*, *Hapavirus*, and *Sunrhavirus*. Most of them were identified in different bird species in Africa, with the exception of Durham virus (DURV), which was isolated from *Fulica americana* in North America in 2005 [11]. This virus is the only bird-related rhabdovirus within the *Tupavirus* genus. Similarly, Landjia hapavirus (LJAV), isolated from *Riparia paludicola* in the Central African Republic in 1970, is the unique representative of bird-related viruses among the *Hapavirus* genus [1]. Conversely, the new taxonomic genus *Sunrhavirus* is associated to a significant degree with birds, and includes Sunguru virus (SUNV), which is described in domestic chicken in Uganda in 2013 [12], Garba virus (GARV), which is reported in *Corythornis cristata* in Bangui in the Central African Republic in 1970 [1], as well as the unclassified rhabdovirus, Farmington virus (FARV), also identified in wild bird in North America in 1969 [13].

In addition to these characterized viruses, other bird rhabdoviruses are still waiting for taxonomic assignation, and include five viruses collected in the Central African Republic with the Bimbo (BBOV), Kolongo (KOLV), Nasoule (NASV), Ouango (OUAV), and Sandjimba (SAJV) viruses, and two rhabdoviruses collected in Egypt called the Matariya (MTYV) and Burg el Arab (BEAV) viruses. At the time of their collection, initial serological analysis evidenced links to rabies virus groups or bovine ephemeral fever ephemerovirus groups [5,14,15]. Subsequent studies based on a phylogenetic analysis of limited regions among the N and L genes repositioned them into the unclassified Sandjimba group [16,17,18,19]. Later, the members of the Sandjimba group, which were most closely related to several bird- or insect-associated rhabdoviruses which originated in America and Africa, were assigned to the new rhabdoviruses genus *Sunrhavirus* [1,12,14,16,17,18,19].

In this study, we extended the knowledge on bird-related rhabdoviruses and their associated genetic diversity through the molecular characterization of the nearly complete genome of these seven unclassified virus species, all of them isolated between 1960s to 1970s in wild birds circulating in Africa. Based on molecular analysis conducted on their genome sequences that were obtained by next generation sequencing (NGS), we confirmed that they belonged to the genus *Sunrhavirus*, sharing a common genomic organization and sequence similarities. Our results demonstrated that these seven bird-related rhabdoviruses could represent putative individual new species which are phylogenetically clustered into a single group within the genus *Sunrhavirus*. These data demonstrate that there is a high level of genetic diversity in the rhabdoviruses that are potentially circulating in African birds. Their close relationship with other insect-related sunrhaviruses, as well as the fact that most of them were isolated from the blood of infected birds, raises questions as to whether they are arthropod-borne and could affect bird communities.

## 2. Materials and Methods

### 2.1. Virus Description

All of the viruses characterized in this study were isolated from wild birds in Africa during large field campaigns that were carried out in the 1960s and the 1970s to collect samples of wildlife, with the main aim of evaluating the presence of arboviruses, in particular by virus isolation and serology characterization. Five of them were collected in the Central African Republic, including Bimbo virus (BBOV, isolate AnB1054d or 9716RCA), Kolongo virus (KOLV, isolate AnB1094d or 9717RCA), Ouango virus (OUAV, isolate AnB1582a or 9718RCA), Sandjimba virus (SAJV, isolate AnB373 or 0408RCA), and Nasoule virus (NASV, isolate AnB4289a or 0410RCA), whereas Matariya virus (MTYV, isolate EgAn1477-61 or 09027EGY) and Burg el Arab virus (BEAV, isolate An3782-62 or 09023EGY) were collected in Egypt [5,15]. All of these samples were collected with nets in various types of landscapes (such as mosaic savanna forests, equatorial humid forests, oases in semi-deserts, or urban/semi-urban areas). The complete description of these viruses is presented in Table 1. Briefly, BBOV and KOLV were obtained from a pool of crushed organs (heart, brain, and spleen) of *Euplectes afer* that were collected in 1970 in the suburb of Bangui city. OUAV and SAJV originated from Landjia city in 1970, and were collected from the blood of *Sitagra melanocephalus* and a pool of crushed organs (heart, brain, spleen, and liver) from *Acrocephalus schoenobaenus*, respectively. NASV was isolated from the blood of *Eurillas virens* (*Andropadus virens*) that were collected in 1970 in Nasoulé city. Lastly, both Egyptian Matariya virus (09023EGY) and Burg el Arab virus (09027EGY) were obtained from the blood of *Sylvia curraca* sampled in Port Saïd in 1961 and in Bahig in 1962, respectively. At the time, all these viruses were isolated, amplified after the intra-cerebral (and sometime intra-peritoneal) inoculation of suckling newborn mice, and stored long term at −80 °C after freeze-drying. The initial characterization of these viruses was performed serologically, using complement fixation, and immunofluorescent and plaque reduction neutralization assays with reference sera panels, leading to classification by serogroup [5,20].

### 2.2. RNA Extraction

Total RNA extraction was performed using Direct-zol™ RNA MiniPrep kit (Zymo Research, Irvine, CA, USA) following the manufacturer’s instructions. Briefly, 600 μL of TRIzol reagent (Invitrogen, llkirch-Graffenstaden, France) was added to nearly 200 μL resuspended lyophilized suckling mice brain in nuclease-free water. A DNase I digestion step was performed on-column, according to the manufacturer’s recommendations. Total RNA extraction was eluted in 50 μL nuclease free and stored at −80 °C for further analysis.

### 2.3. Genome Sequence Determination

Genome sequences of the bird-related rhabdovirus were obtained using next generation sequencing (NGS) as previously described [21,22,23]. A ribosomal RNA depletion step was first carried out with 2−4 μg of RNA with 1 μL of Terminator 5′-Phosphate-Dependent Exonuclease (Epicentre Biotechnologies, Madison, WI, USA), in addition to 2 μL of buffer A and 0.5 μL of RNAsin Ribonuclease inhibitor (Promega, Charbonnières-les-Bains, France). After being adjusted to 20 μL with nuclease-free water, the mix was incubated for 1 h at 30 °C. The depleted RNA was then purified using Agencourt RNAclean XP beads (Beckman Coulter, Villepinte, France) at a ratio of 1:1.8, following the manufacturer’s instructions. Purified RNA was then reverse transcribed in complementary DNA (cDNA) using the Superscript III reverse transcriptase (Invitrogen) according to the manufacturer’s instructions. For this step, 8 μL of RNA was first incubated at 70 °C for 5 min with 1 μL of 10 mM dNTP mix (Invitrogen) and 1 μL of 50 μM of random hexamers (Invitrogen), then placed on ice. The complementary step was performed with the addition of 1 μL (200 U) of Superscript III Reverse transcriptase (Invitrogen), 2 μL of 10 × First-Strand Reaction Buffer, 2 μL of 0.1 DTT, 4 μL of 25 mM MgCl_2_, and 1 μL of RNAsin Ribonuclease inhibitor (Promega) for a final volume of 20 μL. The mix was incubated at 25 °C for 10 min then at 50 °C for 90 min. Afterward, double-stranded DNA (dsDNA) was synthesized in a reaction mix containing 20 μL of fresh cDNA, 10 × Second-Strand Synthesis Reaction Buffer (New England Biolabs, Evry, France), 3 μL of 10 mM dNTP mix (Invitrogen), 1 μL (10 U) of *E. coli* DNA ligase (New England Biolabs), 4 μL (40 U) of *E. coli* DNA polymerase I (New England Biolabs), 1 μL (5 U) of *E. coli* RNase H (New England Biolabs), and 43 μL of nuclease-free water, after incubation at 16 °C for 2 h. The total volume (80 μL) of dsDNA was finally purified for each virus, using a ratio of 1:1.8 of AMPure XP beads (Beckman Coulter) following the manufacturer’s instructions. Finally, dsDNA libraries were constructed using the Nextera XT kit (Illumina, Evry, France) and sequenced using a 2 × 150 nucleotide paired-end strategy on the NextSeq500 platform as previously described [21,22,23].

The NGS data were analyzed using de novo assembly and mapping (both using CLC Assembly Cell, Qiagen, Hilden, Germany) with a dedicated workflow built on the Galaxy platform of Institut Pasteur [21,22,23,24]. Contig sequences were assembled and manually edited to produce the final consensus genome using Sequencher 5.2.4 (Gene Codes Corporation, Ann Arbor, MI, USA). The quality and the accuracy of the final genome sequences were checked after a final mapping step of the original cleaned reads and visualized using Tablet [25].

In case of low covering regions or the presence of gaps, different primer datasets were designed, based either on the sequences obtained by NGS (specific primers) or on the most closely related sequences available in GenBank (Appendix A). These primers were used by conventional PCR (nested or not) using TaKaRa EX Taq (TaKaRa, Kusatsu City, Japan) according to the manufacturer’s instructions, and the amplicons were submitted for Sanger sequencing to fulfil the gaps present in the genome sequences.

Complete genome sequences were deposited in GenBank under the accession numbers MW491754-MW491760.

### 2.4. Sequences Analysis

Identification of putative additional ORFs (≥180 nt) were performed with Sequencher 5.2.4. The presence of accessory genes was evaluated after comparison with similar genome regions of other rhabdoviruses belonging to the same genus which were available in GenBank.

Phylogenetic analysis of the bird-related viruses and other rhabdoviruses was conducted using datasets of reference sequences downloaded from GenBank (Appendix A). Two datasets were used, with the first one including the complete amino acid sequences of the polymerase of 229 other rhabdovirus representatives, whereas the second one encompassed the concatenated nucleotide sequences of the canonical genes (N, P, M, G, and L) of the different sunrhaviruses (14, including the seven new sequences). All these sequences were aligned using ClustalW (version 2.0) [26] or Clustal Omega (version 1.2.4, implemented in SnapGene version 5.3.2) [27], and checked manually for accuracy. Phylogenetic trees were then constructed with IQ-TREE (version 1.6.10) [28] or PhyML (version 3.0) [29], using Maximum Likelihood models with 1000 bootstrap. Sequence identities of were performed using MEGA (version 7.0) [30]. Protein analyses were performed using SnapGene software (version 5.2.3) for predicted molecular weights (MW), isoelectric points (pI), and charge values (at pH 7), using the ProtParam tool on the ExPASy server (https://www.expasy.org/resources/protparam, accessed on 22 September 2021) [31] for the calculation of instability, aliphatic, and grand average of hydropathicity (GRAVY), using TMHMM Server (version 2.0) (http://www.cbs.dtu.dk/services/TMHMM-2.0/, accessed on 22 September 2021) and TMpred server on the ExPASy server to identify transmembrane helices, and using the normal prediction on Phoebius server (https://phobius.sbc.su.se/index.html, accessed on 22 September 2021) [32] to determine the putative protein topology.

## 3. Results

### 3.1. Genome Characterization of the Bird-Related Rhabdoviruses

The determination of the genome sequences of the seven bird-related rhabdoviruses was performed using NGS. Between 0.1 to 12 million raw reads were obtained per sample (around 6 million reads on average) (Table 2). Any remaining gaps and low coverage regions were resolved through specific PCR or nested-PCR and Sanger sequencing of the corresponding amplicons. Each final consensus sequence was then used as a reference sequence for a last mapping round for final verification (Table 2). Nearly complete genome sequences (without the 3′ leader and 5′ trailer sequences) were obtained for all of the seven viruses, which ranged from 10,805 to 11,021 nt in length. The average coverage for each sequence varied from 10x to 600x (Table 2, and Appendix A).

All of the seven bird-related viruses exhibited a typical rhabdovirus genome organization, with the five canonical genes which encode, in the following order, the N (1257−1284 nt, 418−427 aa), P (759−852 nt, 252−283 aa), M (498−522 nt, 165−173 aa), G (1644−1698 nt, 547−565 aa), and the L (6201−6219 nt, 2066−2072 aa) proteins (Figure 1). Numerous putative accessory genes (U) which present additional ORFs were also identified among the genome sequences, from two (with NASV, OUAV, and MTYV) to seven (with BBOV) (Figure 1).

Each of these viruses presented a putative alternative ORF within the P gene (264−312 nt, 87−103 aa), which corresponds to the C protein already found in other rhabdoviruses, such as Sunguru virus (SUNV), Garba virus (GARV), Durham virus (DURV), Tupaia virus (TUPV), or Klamath virus (KLAV). This protein exhibited a variable spectrum of amino acid conservation among the seven newly described bird-related viruses, including GARV (identity 26.1−80.4%), or after comparison with the three other sunrhaviruses, namely SUNV, Harrison Dam (HDV), and Walkabout Creek (WCB) viruses (10.6−28.7%), and they were highly distinct from the other sunrhaviruses and tupaviruses (Appendix A). For all of the seven bird-related viruses, this protein was predicted to be non-polar, approximately neutral, or slightly basic (7.1−8.71) and with a non-cytoplasmic location, similarly to most of the other C proteins found for the sunrhaviruses and tupaviruses, with the notable exception of Klamath virus (KLAV) and TUPV (Appendix A). Another common additional ORF (216−234 nt, 71−77 aa) to all the seven viruses was found between the M and G genes, corresponding to the small hydrophobic (SH) protein previously also observed in the tupaviruses and the previously unclassified SUNV, HDV, WCV, and GARV. Here, again, the identity of the amino-acid sequences of this SH protein was variable but higher between the seven newly described bird-related viruses and GARV (32.4−85.9%) than after comparison to the other sunrhaviruses, namely HDV, SUNV, and WCB (20.2−28.2%), or to the tupaviruses Durham virus (DURV), KLAV, and TUPV (13.8−23.3%) (Appendix A). The SH proteins of the remaining sunrhaviruses Dillard’s Draw virus (DDV), Kwatta virus (KWAV), and Oak Vale virus (OVRV) were clearly distinct from the other SH proteins (Appendix A). The topology of this acidic (4.65−6.55) and hydrophobic protein was similar for the bird-related viruses, with a signal peptide (13−20 aa), followed by an extracellular region (9 or 15 aa), a transmembrane part (18−27 aa) with two amino acid helices, and a cytoplasmic domain (22 or 28 aa) at the C-term (Appendix A). This topology is also observed with SUNV, DDV, and HDV. Lastly, BBOV, KOLV, and BEAV exhibited a putative ORF within the N gene, whereas from one to three additional ORFs were found in the G gene for SJAV and KOLV or for BBOV, respectively. None of these additional ORFs exhibited similarities with other known proteins after a BLASTp analysis of non-redundant protein sequences or Uniprot databases, with default parameters.

The transcription initiation (TI) signal was highly conserved among the five canonical genes, with the AACA sequence motif (Appendix A). The consensus sequence of the transcription termination (TTP) for the canonical genes was TGA_7_, except for the M gene of KOLV and OUAV, which had TGA_6_ and TGA_8_, respectively, or for the L gene of SAJV, which exhibited the CGA_7_ motif. These conserved TI and TTP signal sequences were observed for only one of the putative accessory genes (SH) and for all of the seven bird rhabdoviruses. Surprisingly, the TTP sequence of the SH gene for the virus OUAV was found after the TI sequence of the next gene (G gene) (Appendix A). Intriguingly, a TTP signal sequence was observed just before the TI signal sequence of the N gene for KOLV, suggesting the upstream presence of an additional ORF.

### 3.2. Phylogenetic Analysis of the Bird-Related Rhabdoviruses

A first maximum likelihood phylogenetic analysis was conducted on the complete amino acid sequences of the L protein of the seven bird-related rhabdoviruses, in addition to the 229 representative members of the *Rhabdoviridae* family available in GenBank (Figure 2, and Appendix A). Based on this phylogeny, the seven bird-related rhabdoviruses clustered together into the genus *Sunrhavirus* with high bootstrap supports. Within this genus, they were strongly associated with GARV, one of the other bird-related rhabdoviruses found in this genus, whereas the other bird-related rhabdovirus SUNV was found to be more genetically distant.

A second phylogenetic analysis was conducted on the genus *Sunrhavirus*, based on the concatenated nucleotide sequences of the five canonical genes (N, P, M, G, and L) for the seven different members already associated with this genus, in addition to the seven new sequences (Figure 3). Interestingly, all of these seven bird-related rhabdoviruses clustered into the same phylogroup, identified as Clade III, with GARV, which was found also in birds in Africa, and more precisely in the Central African Republic (Figure 3) [1]. The other unique bird-related rhabdovirus of the genus not related to Clade III was SUNV, which clustered with two Australian insects rhabdoviruses into another clade (Clade II) [17]. However, we observed a close genetic relationship between these two clades (Figure 3). Within Clade III, all the viruses appeared to be genetically relatively distant from each other, suggesting that they could be considered to be individual species.

### 3.3. Genetic Diversity of the Bird-Related Rhabdoviruses

In addition to the phylogenetic analysis, we compared the canonical ORFs (N, P, M, G, and L) of the seven bird-related rhabdoviruses with the other members within the genus *Sunrhavirus*, at the individual level (complete amino acid sequences for each) or after concatenation (concatenated complete nucleotide sequences) (Table 3, and Appendix A). The close genetic relationship between these seven bird-related rhabdoviruses exhibited by the phylogeny was confirmed at the amino acid and nucleotide identity level of these canonical ORFs. These identity analyses also confirmed that these viruses were putative individual species. Indeed, they exhibited a high level of diversity between these viruses and the other members of genus *Sunrhavirus*, with nucleotide identities for the concatenated sequences ranging from 41% to 71.9% among the *Sunrhavirus* genus, and from 55.5% to 71.9% among the seven bird-related rhabdoviruses (Table 3).

At the individual ORF level, the N and L proteins were the most conserved among the member of the *Sunrhavirus* genus, with amino acid identities ranging from 23.8% to 86.6% and 38.0% to 85.2%, respectively (Appendix A). Within the seven newly described bird-related rhabdoviruses, the amino acid identities ranged from 45.3% to 81% and 58.9% to 83.1% for the N and the L proteins, respectively. The level of identity for the amino acid sequences was lower for the other viral proteins among sunrhaviruses, ranging from 5.7% to 71.1%, 7.1% to 79.7%, and 20.3% to 75.5% for the P, M, and G proteins, respectively (Appendix A). For the seven new bird-related rhabdoviruses, identities were 18.6−68.3%, 23.6–66.4%, and 36.1−72.9% for the seven bird-related rhabdoviruses for the P, M, and G proteins, respectively. Altogether, these results indicated that these seven bird-related rhabdoviruses represent new virus species within the *Sunrhavirus* genus.

## 4. Discussion

To date, only 12 rhabdoviruses have been identified in birds. Among them, all except SUNV were collected from wild birds, the latter being isolated from chicken [12]. Two of them originated from the USA, namely DURV and FARV. The other bird-related rhabdoviruses were isolated in Africa and were collected in Uganda (SUNV), Egypt (BEAV and MTYV), and the Central African Republic (BBOV, GARV, KOLV, LJAV, NASV, OUAV, and SJAV) [5,15]. Up to now, complete or nearly complete genome data was available for only four of them, including DURV, which belongs to *Tupavirus* genus, GARV, which is a member of the *Sunrhavirus* genus, LJAV, which clustered with *Hapavirus* genus, and FARV, which remains unassigned [5,15]. For all of the seven other viruses, only partial N or L gene sequences were available, and they were initially classified in the group Sandjimba [1,12,14,16,17,18,19]. In this study, we have enlarged our knowledge about bird-related rhabdoviruses with the determination of the nearly complete genome of these seven other isolates using NGS, which were isolated in various wild birds in Egypt and the Central African Republic of Africa from the 1960s to the 1970s [5,15].

Phylogenetic analysis conducted on the genome sequences demonstrated that they could potentially be all assigned to the newly created *Sunrhavirus* genus, which was set up by the latest update of the International Committee on Taxonomy of Viruses in 2020 (ICTV: https://talk.ictvonline.org/, accessed on 25 September 2021). Seven virus species were associated with this genus, including the two bird-related viruses GARV and SUNV, and five viruses isolated in insects. According to the criteria of the ICTV for species demarcation among the sunrhaviruses, new species need to exhibit a minimum amino acid sequence divergence of 10%, 15%, and 10% for the N, G, and L proteins, respectively. Based on these criteria, all of the seven newly described bird-related rhabdoviruses meet these requirements, with a minimum amino acid sequence divergence of 13.4%, 26.5%, and 16.9% for the N, G, and L proteins, respectively (Appendix A). For these bird-related rhabdoviruses, we therefore propose the tentative name species Sunrhavirus sandjimba for SJAV, Sunrhavirus nasoule for NASV, Sunrhavirus bimbo for BBOV, Sunrhavirus kolongo for KOLV, Sunrhavirus ouango for OUAV, Sunrhavirus Alexandria for BEAV, and Sunrhavirus matariya for MTYV, according to the binomial nomenclature recently adopted for viruses by the ICTV [33].

In addition to the similar length and genetic relatedness of each viral genome between these bird-related rhabdoviruses and the other members of the genus *Sunrhavirus*, other common features were also observed, with the presence of the two accessory genes C (within the P gene) and SH (between the M and the G genes). Similar C proteins can be observed in rhabdovirus genera other than *Sunrhavirus* genus, such as *Vesiculovirus* [34], *Lyssavirus* [35], *Ephemerovius* (with the P’ protein) [36], *Tupavirus*, and *Hapavirus* genera [1], or even in other virus families, such as the *Paramyxoviridae* family. These C proteins, which expression could occur by leaky ribosomal scanning, are generally considered as highly basic. However, this feature was not observed in the case of the newly described bird-related rhaboviruses. Their precise functions remain unknown, although it has been speculated that they could enhance transcriptional activity or act in viral pathogenesis [34]. The small hydrophobic proteins SH are encoded as additional transcriptional units between the M and the G genes for all the sunrhaviruses described so far [1,12,17,19,37]. In addition, similar proteins (coded also as additional transcriptional units between the M and the G genes) have been described for the members of the *Tupavirus* genus, including DURV [11], KLAV [1], and TUPV [38], and more distantly genetically for the members of *Sripuvirus* genus [1]. These proteins are lacking predicted transmembrane domains and presented a highly hydrophilic N-terminal domain, but their function remains unknown [1,3].

Based either on the nucleotide concatenated genome sequences or on the complete amino acid polymerase sequences, all of the seven bird-related rhabdoviruses clustered with GARV in a distinct group, identified here as Clade III, within the genus *Sunrhavirus*. This phylogroup encompasses exclusively viruses from wild African birds (in the Central African Republic and Egypt) [1,5,15]. Two other phylogroups were also observed in this genus, with the phylogroup Clade II closely related to Clade III and encompassing two viruses (HDV and WCV) which were isolated from insects in Australia, in addition to SUNV, which was isolated in Uganda from a chicken [12]. Lastly, the phylogroup Clade I included three sunrhaviruses found in insects of American and Australian origin [1,19,37]. All together, these results demonstrate the high genetic diversity within the *Sunrhavirus* genus, which potentially includes now 14 members from three main geographical areas, namely Africa, America, and Australia.

The fact that the members of the *Sunrhavirus* genus were isolated from insects (mosquitoes with *Culex* species and biting midges with *Culicoides* species) or birds is intriguing, and raises the question about a potential arboviral epidemiological cycle in birds, similar to other bird-related arboviruses, such as West Nile virus [39] or Usutu virus [40]. This is supported by the presence of some of these viruses isolated in the blood of these animals, and that mosquitoes such as *Culex* species feed on these animals [41]. However, studies conducted on SUNV failed to demonstrate virus replication in most of the invertebrate cell lines tested or virus dissemination/infection in the challenged mosquitoes [12]. Similarly, the bird-associated DURV of the *Tupavius* genus was not able to replicate in the insect cell line tested [11]. Further investigations are therefore necessary in order to assess the infection capacities in arthropods of the sunrhaviruses, especially those belonging to the phylogenetic Clade III.

Conversely, it remains also to be elucidated as to what extent these sunrhaviruses could be specifically associated with birds as vertebrate hosts, and what could be the pathologies induced after infection. Indeed, all of the bird-related sunrhaviruses were isolated from animals during nonspecific epidemiological surveillance of arboviruses, without information related to disease or potential clinical signs associated with the infected animals. Only the tupavirus DURV was suspected to be associated with disease in birds, being isolated from a moribund American coot (*Fulica Americana*) that exhibited neurological signs in North Carolina [11]. A first study aiming to evaluate the circulation of HDV and WCV sunrhaviruses in various vertebrates in Australia failed to find serological evidence of infection in wild or domestic birds [17]. However, additional studies, including seroprevalence analysis, will be necessary in order to extend the search for rhabdoviruses in birds in Africa, America, and Australia, but also in other unexplored regions, such as Asia or Europe. Moreover, the possibility of the infection of other vertebrates, such as humans or other mammals, must be considered and evaluated, especially since it has been demonstrated that different mammalian cells can support the replication of some of these viruses, such as SUNV, HDV, OVRV, KWA, and WCV [12,17,19].

Lastly, it is important to underline that the seven bird-related rhabdoviruses described here are part of a large and unique historical collection of arboviruses which were isolated during field campaigns carried out in the 1960s and 1970s on wildlife, particularly in Africa, and of which some of the isolates have yet to be characterized. Thus, complementary studies would make it possible to assess whether these viruses are still circulating and, if so, to investigate their genetic evolution, taking into account changes that may have affected environmental factors or their hosts.

## Figures and Tables

**Figure 1 viruses-13-02168-f001:**
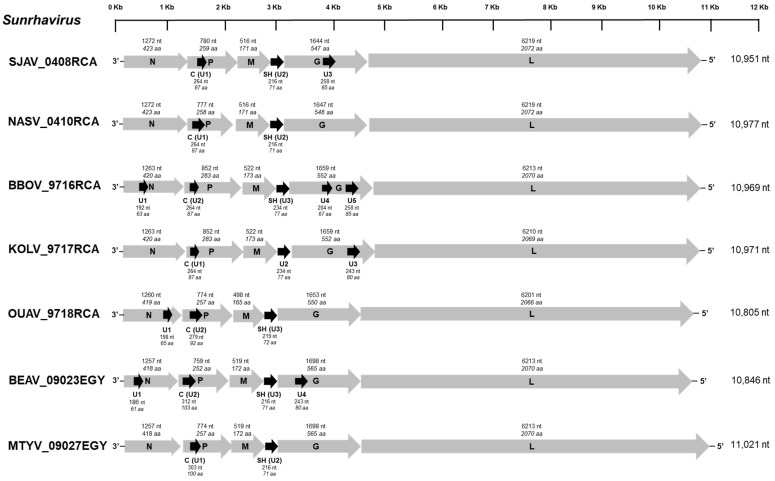
Schematic genome organization of the seven bird-related sunrhaviruses. The gray and black arrows represent the five canonical open reading frames (ORFs) (N, P, M, G, and L), and the putative additional ORFs (≥180 nt), respectively. The length (nucleotide and amino acid) of each ORF is indicated.

**Figure 2 viruses-13-02168-f002:**
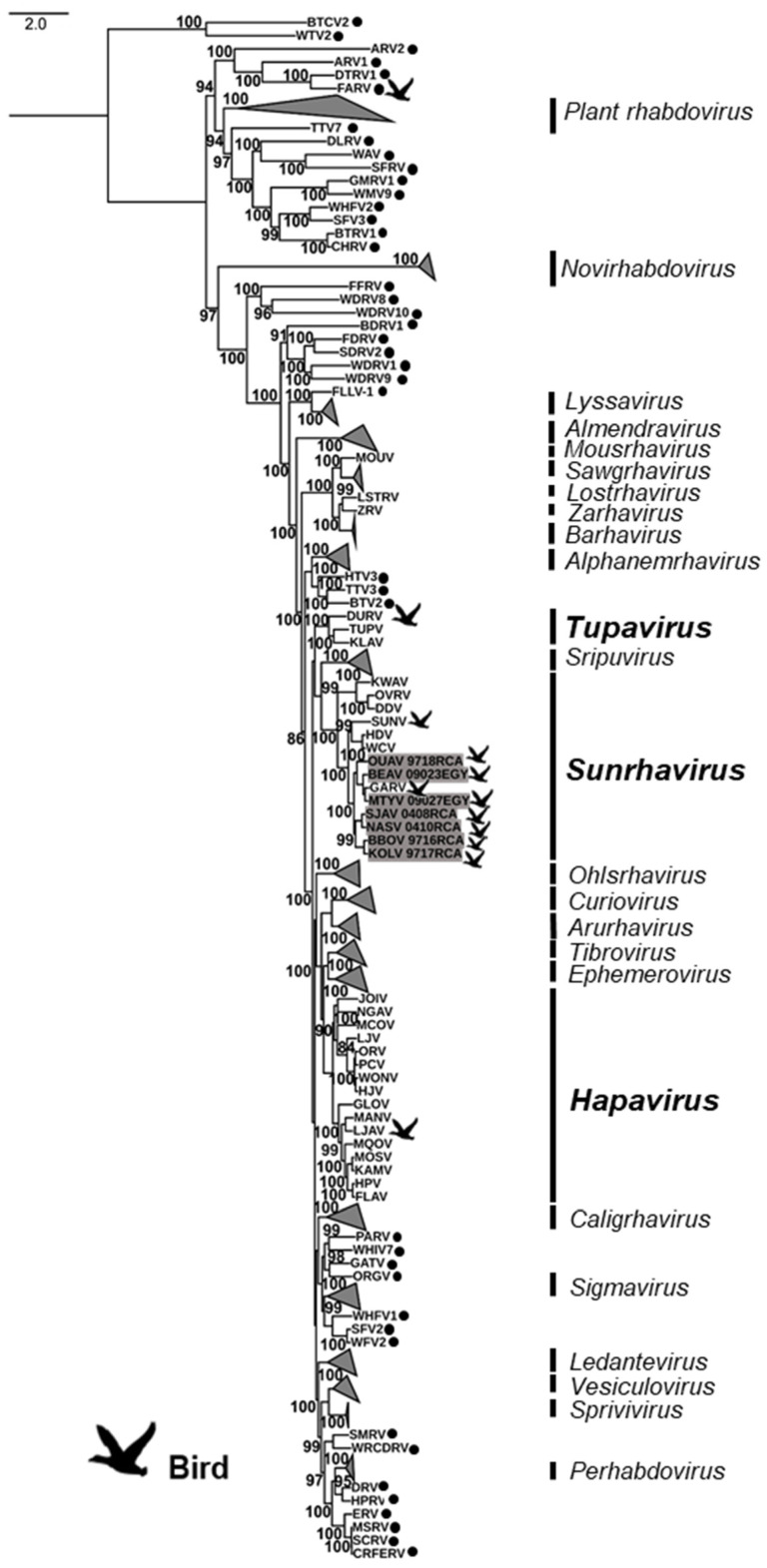
Phylogenetic classification of the seven bird sunrhaviruses. A maximum likelihood phylogenetic tree was made using IQ-TREE1.6.10, on the full amino acid sequence of the L protein, including seven bird-related sunrhaviruses and 229 other rhabdoviruses previously reported on GenBank, using the LG + G + L + F model with 10,000 ultrafast bootstraps. The rhabdovirus genera that were not related to birds were collapsed in the phylogenetic tree. Bird-related rhabdoviruses are indicated by a dedicated symbol, and the bird-related genera are shown in bold. Unclassified rhabdoviruses are indicated by black dots. The bird-related rhabdoviruses described in this study are highlighted in gray. All bootstrap proportion values (BSP) > 80% are specified. The scale bar indicates nucleotide substitutions per site.

**Figure 3 viruses-13-02168-f003:**
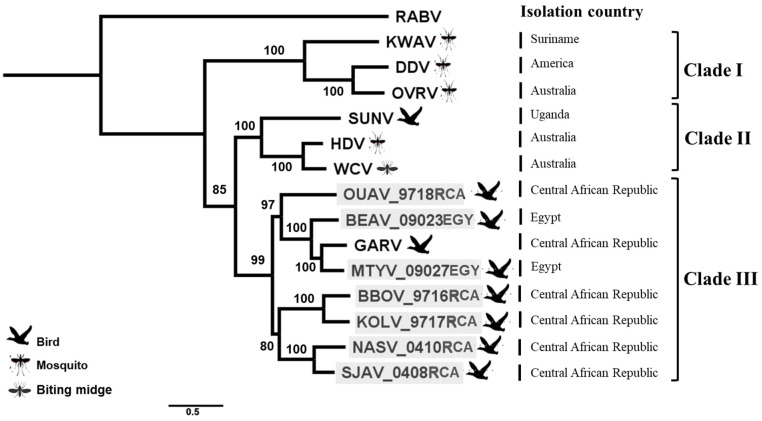
Phylogenetic classification of all members of the genus *Sunrhavirus*, including the seven newly described bird-related sunrhaviruses. A maximum likelihood phylogenetic tree was made with PhyML 3.0 on the nucleotide concatenated ORFs (N-P-M-G-L), using the GTR + G + I model and with 1000 bootstrap replicates. The main animal reservoirs for each virus are indicated by specific cartoons, and the seven bird sunrhaviruses described in this study are highlighted in gray. The isolation country for each virus is presented in the right of the illustration. All bootstrap proportion values (BSP) >80% are specified. The scale bar indicates nucleotide substitutions per site. The classical rabies virus (RABV) was included as outlier in the phylogenetic analysis.

**Table 1 viruses-13-02168-t001:** Description of the bird rhabdoviruses characterized in this study.

Isolate	Name	Acronym	City	Country	Date of Collection	Host Species	Original Samples	Tested Samples
9716RCA	Bimbo virus	BBOV	Kolongo	Central African Republic	18 July 1970	*Euplectes afra*	Unknown	Mouse brain
9717RCA	Kolongo virus	KOLV	Bangui	Central African Republic	18 July 1970	*Euplectes afra*	Pool of crushed organs	Mouse brain
9718RCA	Ouango virus	OUAV	Landjia	Central African Republic	10 October 1970	*Ploceus melanocephalus*	Blood	Mouse brain
0408RCA	Sandjimba virus	SJAV	Landjia	Central African Republic	21 January 1970	*Acrocephalus schoenbaeus*	Pool of crushed organs	Mouse brain
0410RCA	Nasoule virus	NASV	Nasoulé	Central African Republic	15 September 1973	*Andropadus virens*	Blood	Mouse brain
09023EGY	Burg el Arab virus	BEAV	Bahig	Egypt	15 October 1962	*Sylvia curraca*	Unknown	Mouse brain
09027EGY	Matariya virus	MTYV	Port Saïd	Egypt	8 October 1961	*Sylvia curraca*	Unknown	Mouse brain

**Table 2 viruses-13-02168-t002:** NGS results obtained for the genome sequences of the seven bird-related rhabdovirus.

Isolate	Virus	Genome Size (nt)	Raw Reads (no)	Reads Cleaned (no)	Mapped Reads (no)	Average Coverage Depth * (x)	GenBank Accession Number
9716RCA	BBOV	10,969	3,547,036	3,013,558	15,226	206.23	MW491756
9717RCA	KOLV	10,971	4,881,362	4,214,012	46,475	629.49	MW491757
9718RCA	OUAV	10,805	12,256,546	9,754,242	28,150	384.67	MW491758
0408RCA	SJAV	10,951	12,164,616	9,908,488	14,166	191.80	MW491754
0410RCA	NASV	10,977	8,689,566	6,630,310	20,937	281.01	MW491755
09023EGY	BEAV	10,846	101,186	65,258	832	11.15	MW491759
09027EGY	MTYV	11,021	2,742,712	2,289,168	3635	45.58	MW491760

* Sequence coverage obtained after the last mapping round.

**Table 3 viruses-13-02168-t003:** Nucleotide identities of concatenated canonical ORFs (N, P, M, G, and L) of members of the *Sunrhavirus* genus. Identities were calculated through pairwise deletion using MEGA (version 7.0). Newly described bird-related rhabdoviruses are indicated in bold.

	SJAV 0408RCA	NASV 0410RCA	BBOV 9716RCA	KOLV 9717RCA	OUAV 9718RCA	BEAV 09023EGY	MTYV 09027EGY	GARV	HDV	WCV	SUNV	DDV	OVRV	KWAV
**SJAV** **0408RCA**														
**NASV** **0410RCA**	71.9													
**BBOV** **9716RCA**	57.9	56.7												
**KOLV** **9717RCA**	57.9	56.2	71.3											
**OUAV** **9718RCA**	58.4	56.8	56	55.7										
**BEAV** **09023EGY**	57.8	56.3	56.1	55.5	58.4									
**MTYV** **09027EGY**	58.2	56.7	56.4	55.8	58.8	70.1								
**GARV**	58.2	56.5	55.5	55.7	59	69.2	73.2							
**HDV**	51.7	50	50.8	50.5	51.5	51.5	51	51.1						
**WCV**	50.6	48.9	49.8	49.4	50.6	51.3	50.6	50.5	74.8					
**SUNV**	48.9	48.1	47.5	47.2	48.6	48.8	47.9	48.2	54	53.8				
**DDV**	42.4	41.6	42.3	41.8	42.4	42.4	42.3	42.1	43.4	42.5	41.5			
**OVRV**	41	40.5	41	41.5	41	41.8	41.6	41.4	42.4	41.9	40.8	65.2		
**KWAV**	42.3	41.4	41.9	41.7	41.9	41.6	41.8	42	42.7	42.1	41.2	52.1	51.9	

## Data Availability

The data presented in this study are available in the present article and in supplementary material. Complete genome sequences were deposited in GenBank under the accession numbers MW491754-MW491760.

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
