# Peer review of "Genome Characterization of Bird-Related Rhabdoviruses Circulating in Africa"

_viruses, 2021, doi:10.3390/v13112168_

Round 1

Reviewer 1 Report

Dear authors,

sincere compliments for your interesting work.

In my opinion, with regard to materials and methods 2.1 Virus description, you should better explain how the rhabdovirus were identified in the original specimens during the 1960-1970s. References to laboratory techniques are found in introduction (line 83), but this aspect of laboratory practice may be interesting for the reader and briefly discussed.

Furthermore, I focused my attention to the biological specimens you analyzed. I think that the use of 50-60 years old samples may be better explained to an external reader. I can suppose that they represented precious biological material, but they could be also misunderstood and considered a potential bias for the study. Why is it important to exploit this old materials?

In particular, in the results and conclusions you made ecological considerations comparing your studies with recent biological materials. I mean that the hypothesis of a constant environment-host-pathogen system during many years should be better discussed.

Further short annotations:

  • Edit Line 95 “genom e”
  • Line 235-237:figure S4 doesn’t seem to clarify that the SH proteins were distinct from insect SH proteins
  • Line 312-320: with refer to the table S6, the amino acid identity values are different from the values in the text.
    • Protein N 23.8 instead of 23.6%
    • Protein L 38.2 instead of 37.9%
    • Protein P 5.7 instead of 5.1%
    • Protein G 20.3 instead of 19.8%

Author Response

Reply to reviewer 1

Comments and Suggestions for Authors

Dear authors,

sincere compliments for your interesting work.

We would like to thank the Reviewer 2 for his/her time to review our manuscript and to his/her positive feedback.

In my opinion, with regard to materials and methods 2.1 Virus description, you should better explain how the rhabdovirus were identified in the original specimens during the 1960-1970s. References to laboratory techniques are found in introduction (line 83), but this aspect of laboratory practice may be interesting for the reader and briefly discussed.

We thank the Reviewer 1 for her/his useful comment and, as suggested, we added the following description to the Materials and Methods section (2.1):

- Lines 114-117 :”… during large field campaigns carried out in the 1960s and the 1970s to collect samples of wildlife, with the main aim of evaluation the presence of arboviruses, in particular by virus isolation and serology characterization.”

- Lines 122-124: “All these samples were collected with nets in various types of landscapes (such as mosaic savanna forest, equatorial humid forest, oases in semi-desert or urban/semi-urban areas).”

- Lines 135-137: “The initial characterization of these viruses was performed serologically, using complement-fixation, immunofluorescent and plaque reduction neutralization assays with reference sera panels, leading to classification by serogroups [5,20].”

Furthermore, I focused my attention to the biological specimens you analyzed. I think that the use of 50-60 years old samples may be better explained to an external reader. I can suppose that they represented precious biological material, but they could be also misunderstood and considered a potential bias for the study. Why is it important to exploit this old materials?

In particular, in the results and conclusions you made ecological considerations comparing your studies with recent biological materials. I mean that the hypothesis of a constant environment-host-pathogen system during many years should be better discussed.

We would also like to thank the Reviewer 1 for her/his another useful comment. We added the following description to the discussion section:

- Lines 460-466: “Lastly, It is important to underline that the 7 bird-related rhabdoviruses described here are part of a large and unique historical collections of arboviruses which were isolated during field campaigns carried out in the 1960s and 1970s on wildlife, particularly in Africa, and of which some of the isolates have yet to be characterized. Thus, comple-mentary studies would make it possible to assess whether these viruses are still circulating and, if so, to investigate their genetic evolution, taking into account changes that may have affected environmental factors or their hosts.”

Further short annotations:

  • Edit Line 95 “genom e”

Corrected

  • Line 235-237:figure S4 doesn’t seem to clarify that the SH proteins were distinct from insect SH proteins

We corrected the error regarding the figure number (Figure S3 instead of Figure S4). In addition, we added the following sentence in the legend of Figure S3: ”Due to the high genetic diversity, the 3 tupaviruses DUV, KLAV and TUPV compared to the 14 sunrhaviruses, they were aligned separately.”

  • Line 312-320: with refer to the table S6, the amino acid identity values are different from the values in the text.
    • Protein N 23.8 instead of 23.6%
    • Protein L 38.2 instead of 37.9%
    • Protein P 5.7 instead of 5.1%
    • Protein G 20.3 instead of 19.8%

We apologize for these errors, which were fixed

Reviewer 2 Report

The manuscript "Genome characterization of bird-related rhabdoviruses circulating in Africa" reports the analysis of seven archival viruses isolated from bird species. The manuscript is thorough, clear and an addition to Rhabdovirus taxonomy. The only fault is the numerous errors in some sections of the text that require correction or clarification. A list below is provided for the authors.

Line 47. 'diverse'

Line 54. 'near future'

Line 58. '..Klamath virus (KLAV) are related to..'

Line 73. explain 'A the opposite, the new taxonomic..' Do you mean 'Conversely'?

Line 78. 'North America'

Line 90. Clarify the phrase '..we completed the overall knowledge..'

Line 94. 'genome'

Line 100. '..isolated from the blood of..'

Line 101. suggest '..raises questions whether they are arthropod-borne and could affect bird communities.'

Line 258. 'The grey and black arrows..'

Line 288. '..relatively distant..'

Line 338. '..demonstrated that they were assigned to the newly created Sunrhavirus genus..' Is this not something the ICTV decides? Perhaps suggest that they should be assigned to the genus.

Line 357. delete 'different'

Line 360. replace 'which' with 'Such C protein expression could occur..'

Line 379. '..American and Australian origin..'

Line 380. '..which potentially includes now..'

Line 386. 'This is supported by the isolation of some of these viruses from the blood..'

Line 400. '..associated with disease..'

Line 403. '..failed to find serological evidence..'

Line 416.  '.. fulfil gaps in the genome sequences..'

Author Response

Reply to reviewer 2

Comments and Suggestions for Authors

The manuscript "Genome characterization of bird-related rhabdoviruses circulating in Africa" reports the analysis of seven archival viruses isolated from bird species. The manuscript is thorough, clear and an addition to Rhabdovirus taxonomy. The only fault is the numerous errors in some sections of the text that require correction or clarification. A list below is provided for the authors.

We would like to thank the Reviewer 2 for his/her time to review our manuscript and to his/her positive feedback. We proceeded to a deep rereading of our manuscript to fix the different errors in the text and we hope we have been comprehensive for these corrections at this stage.

Line 47. 'diverse'

Corrected

Line 54. 'near future'

Corrected

Line 58. '..Klamath virus (KLAV) are related to..'

Modify as “Tupaia virus (TUPV) and Klamath virus (KLAV) had been related to vesiculovirus…”

Line 73. explain 'A the opposite, the new taxonomic..' Do you mean 'Conversely'?

Yes, exactly. All our apologies for this error.

Line 78. 'North America'

Corrected

Line 90. Clarify the phrase '..we completed the overall knowledge..'

Corrected as follow: “we extended the knowledge on”

Line 94. 'genome'

Corrected

Line 100. '..isolated from the blood of..'

Corrected

Line 101. suggest '..raises questions whether they are arthropod-borne and could affect bird communities.'

Corrected as suggested

Line 258. 'The grey and black arrows..'

It seems that the sentence was already as suggested by the reviewer 2

Line 288. '..relatively distant..'

Corrected line 308

Line 338. '..demonstrated that they were assigned to the newly created Sunrhavirus genus..' Is this not something the ICTV decides? Perhaps suggest that they should be assigned to the genus.

We modified this sentence as follow: “demonstrated that they could potentially be all assigned to the newly created Sunrhavirus genus,”

Line 357. delete 'different'

Corrected

Line 360. replace 'which' with 'Such C protein expression could occur..'

We modified the sentence as follow: “The expression of these C proteins, generally considered as highly basic, could occur by leaky ribosomal scanning, were…”

Line 379. '..American and Australian origin..'

Corrected

Line 380. '..which potentially includes now..'

Corrected

Line 386. 'This is supported by the isolation of some of these viruses from the blood..'

Corrected

Line 400. '..associated with disease..'

Corrected

Line 403. '..failed to find serological evidence..'

Corrected

Line 416.  '.. fulfil gaps in the genome sequences..'

Corrected
